# Play Activities Are Associated with Force Regulation in Primary School

**DOI:** 10.3390/jfmk9010054

**Published:** 2024-03-18

**Authors:** Kyota Koitabashi, An Murase, Jun Yasuda, Takeshi Okamoto

**Affiliations:** 1Graduate School of Health Study, Tokai University, 4-1-1 Kitakaname, Hiratsuka 259-1292, Kanagawa, Japan; 2cgkm002@mail.u-tokai.ac.jp (K.K.); 3cgkm004@mail.u-tokai.ac.jp (A.M.); 2Department of Health Management, School of Health Study, Tokai University, 4-1-1 Kitakaname, Hiratsuka 259-1292, Kanagawa, Japan

**Keywords:** controlled force exertion, play activity, exercise

## Abstract

The daily exercise habits and play activities of children are known to have a significant impact on the development of body control. However, previous studies have not adequately explored the correlation between force regulation during submaximal visual effort, exercise, and play experience. This study aimed to examine the correlation between exercise habits and play experience and their impact on the ability to regulate force. This study involved 23 children with an average age of 9.2 ± 1.0 years. The participants were required to match their force exertion during submaximal effort to a varying demand value displayed in a sinusoidal pattern on a screen (controlled force exertion, CFE). Individual interviews were conducted to gather information on the exercise experience (time, frequency, and duration) and play activities (number of experiences and frequency). Multiple regression analysis was performed to determine the association among exercise experience, play activity, and CFE. The results indicated that the amount of exercise experience was not significantly associated with CFE (β = −0.203, *p* = 0.254). However, in terms of play activities, the number of play experiences was associated with CFE (β = −0.321, *p* = 0.038). On the contrary, play frequency was not significantly associated with CFE (β = −0.219, *p* = 0.191). These findings suggest that play activities are effective in improving force regulation during childhood and that a greater variety of play experiences may be important.

## 1. Introduction

Habitual physical activity and outdoor play are associated with decreased sedentary behavior and higher moderate-to-vigorous physical activity in children [1]. In addition, active play has been proposed as a beneficial approach for enhancing children’s fundamental motor skills [2]. Consequently, active physical activity in children may lead to positive outcomes for motor skill development and health.

In recent years, there has been growing concern regarding the occurrence of injuries among children in Japan. According to a report by the Japan Center for the Promotion of Sports [3], approximately 70% of injuries in elementary school children are associated with the upper limbs, and approximately 30% of these injuries are related to the head and face. A meta-analysis explored the causes of facial trauma in children and adolescents, identifying “falls” and “during sports” as common factors [4]. These injuries may be attributed to a lack of proficiency in body control and the inability to effectively avoid falls in hazardous situations encountered in daily life and during physical activities.

When encountering hazardous situations, the ability to control the body and react quickly is important to avoid falls and injuries. These abilities are known as agility [5] and motor coordination, which involve controlling multiple body parts and moving the entire body [6]. Exercise and play facilitate the development of body control abilities in children. Previous studies have shown that continuous sports club participation during childhood enhances motor coordination [7]. Children who engage in multiple sports training programs gain better motor coordination than those who participate in a single sport [8]. Similarly, children who preferred dynamic play activities demonstrated improved agility [9]. Engaging in playground activities involving various types of equipment can enhance motor coordination [10]. Therefore, it can be inferred that the daily exercise habits and play activities of children have a significant impact on the development of body control.

Previous research has primarily focused on assessing the outcomes of maximum skill execution (e.g., time, distance, number of times) when studying body control. There has been a growing trend in studies on children’s motor coordination. The assessment tools commonly used in these studies, such as the Körperkoordinationstest für Kinder and the side-to-side jump, have gained widespread popularity [8,11,12,13,14]. These assessment tests typically measure motor performance during maximal effort with the goal of executing a prescribed exercise in the shortest possible time, maximum number of times, or maximum distance. However, in real-life situations, the ability to appropriately regulate force based on the intended purpose is crucial. Therefore, it is important to shift focus toward purposeful force regulation during submaximal effort.

Research has been conducted to investigate force regulation during tasks that require submaximal effort with a focus on meeting a specified target value. Ohtaka et al. [15] conducted a study involving grasping, jumping, and throwing tasks with elementary school children, in which they were instructed to exert half the force based on their subjective sensation of maximal effort. However, in daily life, force adjustment often relies on visually perceived information. For example, visual input plays an important role in perceiving the shape and context of complex objects when handling them [16]. CFE is the ability of the nervous system to adjust force levels to minimize discrepancy between the required and exerted values while exerting muscular force in response to fluctuating demand values. Nagasawa et al. developed a controlled force exertion (CFE) test to assess this [17]. This test involves coordinating force exertion to match a visually presented target using the grasping motion and grip strength. The required force values are presented as bar graphs or waveforms, allowing for an objective evaluation of force regulation. Therefore, when examining force regulation during submaximal effort, it is important to not only consider subjective force exertion, but also visually assess the task of adjusting to the required force.

Previous studies have not adequately explored the correlation between force regulation during submaximal visual effort, exercise, and play experience. Although force regulation during submaximal effort in adolescence has been suggested to be linked to locomotion during the ages of 7–9, this finding is based on a retrospective study and subjective measures of force exertion [18]. Therefore, investigating the association between exercise/play and force regulation in children is urgently needed. Therefore, this study aimed to elucidate the relationship between force regulation during submaximal visual effort, exercise, and play experiences in children.

## 2. Materials and Methods

### 2.1. Study Design

This cross-sectional study was conducted between April and May 2023.

### 2.2. Participants

To participate in this study, an invitation for research participants was extended to specific sports clubs. Participants were eligible for inclusion if they were registered members of the specified sports club, maintained regular attendance, were in good physical condition, and did not have any wrist injuries. Individuals who were not maintained regular attendance with a sports club or encountered difficulties undergoing measurements due to injury or other factors were excluded from the study.

The research consisted of 23, only boys, with an average age of 9.2 ± 1.0 years, who were actively participating in a sports club located in K prefecture. Sports clubs offer multiple sports, exercise and play programs aimed at improving basic physical fitness and motor skills. The club is comprised of children in various grades, spanning from first to sixth grade, including those with prior experience in sports at other sports clubs. The participants in this study were habituated to regular exercise and play activities. The age distribution of the participants in the study is presented in Table 1. Regarding club membership, 5 children were exclusively affiliated with this sports club, 14 children were members of this sports club and another, and 4 children were members of this sports club as well as 2 other sports clubs (Table 2).

The dominant hand of each child was determined based on their preference for daily activities, including writing, using chopsticks, and throwing balls. All participants were assessed and found to be in optimal health without any reported wrist injuries.

The study’s objectives and research methods were verbally communicated to both club representatives and the children involved. The parents of the children were provided with written and video information, and their consent to participate in the study was obtained using a consent form. Once informed consent was obtained from both the children and their legal guardians, they were provided with the option to willingly participate in the study. This study was conducted in accordance with the Declaration of Helsinki and approved by the Tokai University Ethics Committee on Research Involving Human Subjects (23004).

### 2.3. Measures and Procedure

The objective of this study was to evaluate the maximum muscle strength (maximum grip strength) and coordination ability of muscle strength through grasping movements. Individual interviews were conducted to gather information on the exercise experience and play activities.

#### 2.3.1. Measurement Device

The study employed a digital grip dynamometer (T.K.K 5710b, Takei Kiki Kogyo, Niigata, Japan) equipped with a strain gauge to assess maximum grip strength and CFE. The grip force values were converted into electrical quantities and subsequently recorded using a strain-up device (TSA-210) at a sampling rate of 10 Hz. The TSA-210 strain-up device was connected to a personal computer using a USB cable, and the measurement screen was presented on a monitor located 50 cm from the subject.

#### 2.3.2. Grip Strength

The participants were instructed to sit on a chair and place their elbows on a table. Each participant calibrated a grip strength meter to ensure that the second joint of the hand was flexed at a 90° angle. The maximum grip strength was measured twice using the dominant hand. A 1 min interval was allocated between measurements.

#### 2.3.3. CFE Test (Figure 1)

In this study, the measurement procedure was conducted by referring to the methodology established in previous studies [17]. Participants were instructed to assume the same posture during the assessment of maximum grip strength. A sine wave, ranging from 5% to 25% of the individual’s maximum muscle strength, was presented on a screen with a frequency of 0.1 Hz, and used as the designated target value. Participants were instructed to exert grip strength to minimize the disparity between their measured grip strength and the specified value indicated by the sine wave. The actual grip strength was visually and spatially depicted as a temporal progression from left to right. Participants were instructed to hold the grip strength meter in a manner that corresponded to the alignment of the red line. The experimental protocol comprised a single practice trial, followed by two main trials, with a minimum rest period of 1 min between each trial. Each trial lasted 45 s, with the first 5 s excluded from the analysis. The evaluation variable employed in this study was the absolute sum of the discrepancy between the required and actual grip strength, which was quantified as a percentage. CFE was determined by selecting the trial with the lowest total error percentage between the second and third trials.

**Figure 1 jfmk-09-00054-f001:**
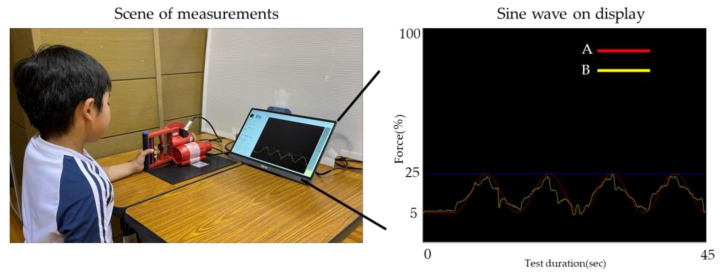
Sine wave A (red line) shows the demand value, while sine wave B (yellow line) shows the actual exertion of grip strength. The test was to match sine wave B with A, which varied in the range of 5% to 25% of maximal grip strength value. The test time was 45 s for each trial. The controlled force exertion was calculated using the data from the 40 s of the trial following the initial 5 s of the 45 s period.

#### 2.3.4. Exercise Experience

Exercise is a physical activity that is planned, structured, and repetitive to improve and maintain physical fitness [19]. Regarding exercise experience, we define the term “exercise experience” as the physical activity conducted in a sports club. We gathered data on the duration of each exercise session (min), frequency of exercise per week (weeks), and duration of exercise (months). This information was collected from activities carried out at the children’s sports club and those conducted at other sports clubs for exercise and sports. To determine the exercise experience of the participants, we divided the total number of days in 1 year (365) by seven, resulting in the calculation of the number of weeks in 1 year (52.14). We divided 52.14 by 12 to determine the number of weeks in 1 month, which was 4.35. The formula for calculating the total exercise experience was as follows:total time = time per exercise session (min) × frequency per week (weeks) × 4.35 × duration of exercise (months).

The cumulative total amount of exercise was determined by aggregating all individual exercise experiences.

#### 2.3.5. Play Activity

Active play is characterized by increased energy, exerted, rough, tumble, gross motor, unstructured, free choice, and fun [20]. In this study, we define the term “Play activity” as the physical activity (tag, playground equipment usage, ball, etc.) that includes the above elements, excluding school physical education classes and the sports club. Regarding such play activity, the study gathered data on the frequency (times per week) and number of experiences. The number of experiences refers to how many types of play activities individuals engage in, which was determined based on the number of reported types of play (e.g., tag, soccer, rope: 3 types of play). The variables employed for evaluating play activities included current play frequency (times per week) and number of play experiences.

### 2.4. Data Analysis

The factors encompassed various aspects, such as maximum grip strength, CFE, exercise experience at the subject’s sports club, exercise experience at other sports clubs, total exercise experience, play frequency, and number of play experiences. Descriptive statistics including means and standard deviations were computed for each variable. Multiple regression analysis was performed to examine the association between the total quantity of exercise experiences, play activity, and CFE. The analysis incorporated age and maximum grip strength as independent variables, with each variable (total exercise experience, number of play experiences, and play frequency) entered into a distinct model. We calculated the effect size (Cohen f^2^) for each model in the multiple regression analysis. The standard definitions of Cohen’s f^2^ for multiple regression are as follows: small, 0.02; medium, 0.50; large, 0.80 [21]. We used statical power program (G*Power v.3.1.9.7; Heinrich-Heine-Universität, Düsseldorf, Germany). Statistical analyses were performed using the SPSS software (IBM SPSS Statistics Version 26, SPSS Inc., Chicago, IL, USA). The statistical significance level for all tests was set at *p* < 0.05.

## 3. Results

The mean values, standard deviations, range, minimum value, maximum value of the maximum grip strength, CFE, time, frequency, duration, and total time of exercise experience at the sports clubs and other sports club attended by the children in the study, and the total quantity of exercise experiences obtained by combining all exercise experiences, number of play experiences, and play frequency are displayed in Table 3.

In this study, a multiple regression analysis was conducted to examine the association between total exercise experience, play activity, and CFE. The findings, as shown in Table 4, demonstrate that all models displayed a notable inverse relationship with maximum grip strength. Model 1 did not exhibit a statistically significant association with the total quantity of exercise experiences (β = −0.203, *p* = 0.254). The adjusted R^2^ indicates that approximately 47% of the variance could be accounted for, with a large effect size (Cohen’s f^2^ = 0.88), while Model 2 exhibited a statistically significant inverse relationship with the number of play experiences (β = −0.321, *p* = 0.038). The adjusted R^2^ indicates that approximately 55% of the variance could be accounted for, with a large effect size (Cohen’s f^2^ = 1.21). Model 3 did not exhibit a statistically significant association with the frequency of play activity (β = −0.219, *p* = 0.191). The adjusted R^2^ indicates that approximately 48% of the variance could be accounted for, with a large effect size (Cohen’s f^2^ = 0.90). To evaluate the existence of multicollinearity, this study employed variance inflation factor (VIF) values. None of the VIF values exceeded 10 in any of the models, thereby confirming the absence of multicollinearity.

Furthermore, for total quantity of exercise experience (TQEE), we divided the participants into 3 quantiles (TQEE Low [*n* = 8], TQEE Middle [*n* = 8], TQEE Large [*n* = 7]). Analysis of covariance (ANCOVA) was finally carried out to analysis the influence of TQEE on CFE (Appendix A).

## 4. Discussion

The primary objective of this study was to investigate the association among exercise experience, play activity, and force regulation in children. The findings revealed that the number of play experiences was significantly associated with force regulation. However, no statistically significant effect was observed on the total amount of exercise experienced in sports club activities. To the best of our knowledge, this is the first study to propose that play activity influences force regulation during submaximal effort.

A significant correlation was identified between the number of play experiences and CFE. Previous studies have demonstrated that the implementation of play interventions and outdoor play can improve children’s body control abilities [22]. Children who engaged in regular playground activities weekly for 10 weeks demonstrated significant improvements in overall motor coordination [10]. Additionally, participation in target play twice a week for approximately one month, along with horizontal-bar play, significantly improved force coordination in young children during throwing movements [23,24]. The evidence suggests that engaging in outdoor activities significantly affects the development of fundamental motor skills. This emphasizes the importance of participating in outdoor activities [25]. This study contributes substantially to the existing literature by proposing that the amount of play experience, specifically the number of experiences, is significantly linked to force regulation during submaximal effort. These findings underscore the potential advantages of participating in various play activities for improving force regulation during submaximal effort. Consequently, these findings indicate that engaging in play activities during childhood may serve as an effective strategy for enhancing motor coordination and force regulation during submaximal effort.

In this study, we observed no significant association between the degree of exercise experience and CFE. Previous studies have indicated that involvement in multiple sports and a longer duration of participation may positively impact motor coordination [8,26]. However, there are variations in motor coordination among athletes, with their level of proficiency in specific tasks contingent on the specific sport in which they are involved [27]. This suggests that specialized training programs designed to meet the specific requirements of each sport can enhance motor coordination by focusing on sports-specific movements and intensities. The participants in this study may have encountered different conditions during their usual sporting activities, as they were required to regulate force through grasping movements, which accounted for approximately 25% of their overall performance.

This study revealed a significant association between the ability to exert maximum effort and the CFE on grip strength. Prior studies investigating force regulation during maximal effort have demonstrated no significant correlation between maximal effort and force regulation in children aged 5–11 years [15,28]. In this study, a correlation was observed between maximal exertion and the ability to regulate force and grip strength, which contradicts previous research findings. Notably, although force regulation was comparable, the previous study involved a task that relied on subjective sensation, whereas the current study employed a task that relied on visual information, thereby representing a distinct condition. This observation is consistent with the results of previous studies that have identified a correlation between visual tasks, maximal muscle strength, and force regulation in children [29]. The findings of this study indicate that the significance of maximal muscle strength may differ depending on the specific circumstances of the task.

### Limitations

This study has some limitations. First, the examination of the association between force regulation development, exercise experiences, and play activities was not feasible because of the inclusion of only male participants and the limited number of subjects within the same age group. Future research should gather longitudinal data on the development of force coordination, comprehensive exercise experiences, and play activities. Second, in participants for the study, sampling bias may have occurred. The participants were children affiliated with sports clubs, which potentially indicates a high level of physical activity. Furthermore, the inclusion of only male participants raises questions regarding the applicability of the findings to a broader population of children. To investigate the effects of different exercises and play activities on force regulation, it is necessary to broaden the study to include a wider population of children without established exercise habits and including girls. Third, the results of this study may be specific to the Japanese population and may not be easily extrapolated to other cultural and demographic contexts. Fourth, this study employed a grasping movement test to assess the muscle strength coordination ability as a measure of force regulation. Therefore, it is important to conduct a more comprehensive evaluation to determine whether the task used in this study effectively captures the children’s ability to regulate force, including other movements and various types of force regulation tasks. Also, we investigated the dominant hand using original items. In future studies, involving assessments of grasping movements, such as in this study, the dominant hand should be examined using a standardized questionnaire.

## 5. Conclusions

This study investigated the association between exercise and play on the CFE in male children. The findings revealed that play activities are associated with force regulation. The number of play experiences influenced force regulation. These findings emphasize the importance of play activities as essential physical pursuits for improving force regulation during childhood. They also highlight the significance of including diverse play activities. Consequently, it is suggested that play activities are one of the important physical activities, and it is advisable to expose children to unstructured play activities to enhance their physical functioning.

## Figures and Tables

**Table 1 jfmk-09-00054-t001:** Profile of subjects (*n*= 23).

Age Category	7 Years	8 Years	9 Years	10 Years	11 Years
*n* (%)	1 (4.3)	3 (13.0)	13 (56.5)	2 (8.7)	4 (17.4)

**Table 2 jfmk-09-00054-t002:** Number of sports club affiliations.

Sports Club Affiliations	*n* (%)
One	5 (21.7)
Two	14 (60.7)
Three	4 (17.4)

**Table 3 jfmk-09-00054-t003:** Characteristics of subjects (*n* = 23).

		Mean	SD	Range	Min	Max
Maximum grip strength (kg)		17.5	4.6	19.3	9.2	28.5
CFE (%)		1161.2	395.7	1828.4	588.3	2416.7
Activities at the sports club (*n* = 23)	Time (min/day)	92.6	34.9	70	50	120
	Frequency (times per week)	1.5	0.8	2	1	3
	Duration (month)	26.7	15.7	59	1	60
	Total time (min) ^a^	20,230.9	22,329.6	93,751.2	208.8	93,960
Activities at other sports club (*n* = 18)	Time (min/day)	150.3	83.0	270	60	330
	Frequency (times per week)	2.6	1.5	6	1	7
	Duration (month)	41.4	27.4	47	1	48
	Total time (min)	68,274.1	80,230.7	237,029.8	1002.2	238,032
Total quantity of exercise experiences (*n* = 23) ^b^		73,662.8	85,330.5	287,935.2	208.8	288,144.0
Play activity (*n* = 23)	Frequency (times per week)	4.1	1.5	6	1	7
	Number of experiences	2.5	1.0	4	1	5

^a^ total time: time × frequency × 4.35 × duration. ^b^ total quantity of exercise experiences: total time at the sports club + total time at other sports clubs. SD, standard deviation; CFE, controlled force exertion.

**Table 4 jfmk-09-00054-t004:** Association of CFE with dependent variables.

	Variable	β	*p*-Value	Adjusted R^2^
Model 1	Age	0.482	0.05	0.469
	Maximum grip strength	−0.928	<0.001	
	Total quantity of exercise experiences ^a^	−0.203	0.254	
Model 2	Age	0.374	0.081	0.549
	Maximum grip strength	−0.933	<0.001	
	Play activity (number of experiences)	−0.321	0.038	
Model 3	Age	0.329	0.158	0.481
	Maximum grip strength	−0.930	<0.001	
	Play activity (frequency)	−0.219	0.191	

^a^ total quantity of exercise experiences: total time at the sports club + total time at other sports clubs. Multiple regression analysis was performed with CFE as the dependent variable and age and maximum grip strength as the independent variables, with each variable (total quantity of exercise experiences, number of play experiences, play frequency) being entered as a distinct model. CFE, controlled force exertion.

## Data Availability

The data presented in this study are available on request from the corresponding author. The data are not publicly available due to ethical restrictions.

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
