# Peer review of "Play Activities Are Associated with Force Regulation in Primary School"

_jfmk, 2024, doi:10.3390/jfmk9010054_

Round 1

Reviewer 1 Report

Comments and Suggestions for Authors

I appreciate the opportunity to review this interesting brief report about how are play activities associated with force regulation in primary school children in which the authors have concluded from models regressed on data provided through a cross-sectional design. Even though I find this manuscript interesting, some serious methodological questions arose upon the manuscript reading which I will address below in a distinct section, after the point-by-point minor revision.

Minor revision

Please include line numbering to facilitate a review process.

BACKGROUND

1.       [First paragraph] I find such extensive information on the injury incidence redundant since it is unrelated to the topic; thus, no risk of injury has been estimated in this study. Moreover, the authors have not provided a reference on how are force regulation (body control?) and risk of injury (falls) related. Please make sure that the study rationale is genuine, direct, and concise and that matches the study outcome.

2.       [Second paragraph] I cannot comprehend how force regulation, body control, motor coordination, and agility correlate; have the authors used them as synonyms or something else? What is force regulation/force adjustment? Please provide a theoretical framework and define them.

3.       [Second paragraph] Please also provide a framework for the types of sports being studied to match the studied factors (included independent variables) – e.g. multisports vs single sport, dynamic vs static sport, with/without the equipment.

4.       [Second paragraph] Multisport activities can be incorporated into a single club rather than participating in several sports clubs, and such activity (multisport) is very popular. See the article 10.3390/ijerph17165902 Moreover, multisports vs single sport shouts another well-known principle: LTAD - Long-term athletic development  10.1080/02640414.2019.1647032   

5.       [Second paragraph] Were the authors keen to explore the role of daily exercise habits and play activities as habitual and structured physical activity, respectively? Please explain and define it pursuant to the studied factors.

6.       [Third paragraph] How are body control and “quickness of movement” related? What is “quickness of movement” and why is it under the question marks? Please provide the framework and thus reference (same as 2).

7.       [Third paragraph] I am not convinced that the KTK performance depends on the time needed to finish the tests. Please provide the reference and explain why is it important to appropriately regulate force during submaximal effort in primary school children.

8.       [Fourth paragraph] What did Ohtaka et al. conclude in their study?

9.       [Fourth paragraph] Please reference that the force adjustment relies on the visual input.

10.   [Fifth paragraph] Why did the authors find that the previous studies were inadequate, and how is this study superb in comparison to those retrospective studies? Is the level of evidence higher? Please explain.

MATERIALS AND METHODS AND RESULTS

11.   [Participants, First paragraph] Please state that only boys participated in this study.

12.   [Participants, First paragraph] Have the authors estimated the minimum sample required?

13.   [Participants, First paragraph] I find the descriptive statistics in the text over extensive and suggest presenting the data tabularly.   

14.   Participants, [First paragraph] Please include inclusion and exclusion criteria for the study participants.

15.   [Participants, Second paragraph] Did the authors use a standardized questionnaire to determine a hand preference such as the Edinburgh Handedness Inventory?

16.   Please reference and provide the reliability and validity data for the CFE test.

17.   Please reference the interpretation of the effect size measure.

18.   I suggest merging Tables 2 and 3.

Major revision

19.   I find that the measurements of exercise habits and play activities are not defined in the manuscript or standardized, and hence, the variables are not scaled. What does the variable Play activities represent? I suggest reporting descriptive statistics such as range, min, max, and, if possible, a reference. I find this as a serious methodological issue. Also, could variables be differently binary or ordinary coded, e.g. PA (0 = one sport, 1 = more than one sport)?

20.   Please report whether the assumptions are true for using regression analysis, such as the existence of outliers and leverages, the normality of residuals, correlations among the outcome (force regulation, dependent variable) and factors (age, grip strength, EH, PA = independent variables), etc..

21.   I also suggest modeling a force regulation in the function of each variable in blocks with each block representing one factor. Maybe there are interactions among the factors, have the authors tested them? Also, have the authors tested are the relationships linear between the outcome and the factor in nature, or could polynomials fit the relationships better? I suggest testing various models to find the best fit.

22.   Upon visual inspection of Table 3 in the manuscript I found an issue with the variable Total time because SD exceeds AS which violates the normality assumption. Please explain.

23.   I find the conclusion about the influence of play activities on force regulation (hence the study title) could not be supported by this study data since no controls (no play activity) were included, but rather a conclusion about the influence of the multilateral sports experience (LTAD) and the time spent physically active on force regulation in boys. I suggest redefining it. Please explain.

Finally, the review stops here due to the urgent authors’ need for subsequential changes to be made. 

Reviewer 2 Report

Comments and Suggestions for Authors

Rewiew

The research presented in this study investigates the correlation between daily exercise habits, play activities, and their impact on the ability of children to regulate force during submaximal visual effort. While the link between physical activity and overall development in children is well-established, the specific relationship between force regulation and exercise/play experiences has not been thoroughly explored in previous studies.

Understanding the factors that contribute to the development of body control in children is crucial in contemporary times due to several reasons. Firstly, there is a growing concern about sedentary lifestyles and reduced physical activity among children, which has been associated with various health issues such as obesity and related conditions. Exploring the nuances of how different types of physical activities, including play, influence specific aspects of development, such as force regulation, provides valuable insights for promoting healthier and more active lifestyles among children.

Secondly, as technology continues to play a significant role in children's daily lives, investigating the impact of submaximal visual effort on force regulation becomes relevant. With the prevalence of screen-based activities, understanding how these activities influence motor skills development can have implications for designing interventions to mitigate potential negative effects.

The study's findings suggest that while the amount of exercise experience did not show a significant association with force regulation, the number of play experiences was correlated with improved force regulation during submaximal visual effort. This highlights the importance of a diverse range of play activities in enhancing specific aspects of motor skills in children.

The research contributes to the current understanding of the relationship between physical activities, play experiences, and force regulation in children. The findings underscore the importance of promoting a variety of play activities to positively impact the development of body control in children, providing valuable insights for educators, parents, and health professionals aiming to enhance the well-being of the younger generation in the context of evolving lifestyle patterns.

some suggestions:

The main criticism of this research could be related to the generalization of findings and the specificity of the study population. The introduction highlights the concern about injuries among children in Japan, particularly those associated with the upper limbs and head/face during sports and falls. While the context is relevant, it is crucial to recognize that the study's findings might be specific to the Japanese population and may not be easily extrapolated to other cultural or demographic contexts.- This should be emphasized in the limitation chapter!

Another potential criticism may be related to the reliance on self-reported data for exercise habits and play activities. The study gathered information on the duration, frequency, and type of exercise and play from the participants and did not employ objective measures. Critics may argue that self-reported data can be subject to recall bias or social desirability bias, leading to potential inaccuracies in the reported exercise habits and play activities. The study could have benefitted from objective measurements or additional verification methods to enhance the reliability of the gathered data.- This should be emphasized in the limitation chapter too!

One potential criticism of the discussion in this scientific research is related to the generalization of findings and the limitations of the study.

Limited Generalizability: The study primarily includes male participants affiliated with sports clubs, potentially indicating a high level of physical activity. This limited and specific sample raises concerns about the generalizability of the findings to a broader population of children without established exercise habits or different demographic backgrounds. The results may not be applicable to females or children who are not part of sports clubs, limiting the external validity of the study.

Sampling Bias: The discussion acknowledges that the study included only male participants, and it points out the need for future research to include a wider population of children. However, the limitations section does not address this potential sampling bias comprehensively. –Please write about in the limitation!

It deals with a very important topic, after developing minor amendment proposals and more detailed limitations, I will professionally support its publication.

Round 2

Reviewer 1 Report

Comments and Suggestions for Authors

I appreciate the authors' commitment to fulfilling the needs; the manuscript and the result sections sound more apparent to readers. However, I must point out a data analysis issue through the comments below.  

1-14 Comments: I appreciate the authors' effort to answer the questions; hence, I have nothing more to add.   

Comment 15: Please include it as a study limitation.

Comment 17: I cannot find a reference for ES in the data analysis section. Please provide one.

Comment 19: Thank you for the response. I now understand what the variable represents. However, the variable Playtime has SD>Mean, which might indicate that the data is overdispersed and/or heavy-tailed and that the values are far from the mean; hence, Playtime's mean poorly represents the sample. Specifically, it could indicate heteroscedasticity, where the variability of the residuals (the differences between observed and predicted values) changes as the value of the independent variable changes - this violates one of the critical assumptions of linear regression, which assumes constant variance of residuals across all independent variable levels. Therefore, it's essential to consider potential transformations or alternative modeling approaches to address this issue, such as using generalized linear models or transforming the variables to achieve constant variance. Please refer to the comment 22.

Comment 20: Thank you for the response. The KS test should be performed on the residuals of each predictor in the regression model (thus, not only one p but for each predictor). Moreover, the KS test is not the only assumption for testing a linear regression model. Please check this: https://doi.org/10.1111/ceo.12358 or similar articles and report accordingly.

Comment 21: Thank you for the response. Nevertheless, regarding the researchers' goal https://doi.org/10.1016/j.pmrj.2014.08.941, how did the authors intend to explore the relationship, if not exploratory? What I meant in the previous comment is not to build a predictive model but to examine the nature of the relationship by finding the best-fit model.  

Comment 22: Thank you for the response. I believe that the calculations were valid. I intended to ask to transform the continuous variable to a categorical one to ensure a known distribution. I suggest equal intervals, quantiles, cluster analysis, or a similar approach to group the data - this may, in turn, allow testing of the effects of time spent exercising as a factor. Also, the authors could create a variable accompanying whether a child participated in one or more clubs (e.g., 0 – one, 1 – more than one) and test it in the model—the same stands for the CFE since the normality was violated.
